# Genetic continuity, isolation, and gene flow in Stone Age Central and Eastern Europe

Tiina M. Mattila [1✉], Emma M. Svensson[1], Anna Juras [2], Torsten Günther[1], Natalija Kashuba [1,3], Terhi Ala-Hulkko [4,5], Maciej Chyleński [2], James McKenna[1], Łukasz Pospieszny[6,7], Mihai Constantinescu[8,9], Mihai Rotea[10], Nona Palincaş [11], Stanisław Wilk[12,13], Lech Czerniak [7], Janusz Kruk[14], Jerzy Łapo[15], Przemysław Makarowicz [16], Inna Potekhina[17,18], Andrei Soficaru[8], Marzena Szmyt[16,19], Krzysztof Szostek[20], Anders Götherström[21,22], Jan Storå[22], Mihai G. Netea [23,24], Alexey G. Nikitin [25], Per Persson [1,26], Helena Malmström [1,27] & Mattias Jakobsson [1,27,28✉]

The genomic landscape of Stone Age Europe was shaped by multiple migratory waves and population replacements, but different regions do not all show similar patterns. To refine our understanding of the population dynamics before and after the dawn of the Neolithic, we generated and analyzed genomic sequence data from human remains of 56 individuals from the Mesolithic, Neolithic, and Eneolithic across Central and Eastern Europe. We found that Mesolithic European populations formed a geographically widespread isolation-by-distance zone ranging from Central Europe to Siberia, which was already established 10,000 years ago. We found contrasting patterns of population continuity during the Neolithic transition: people around the lower Dnipro Valley region, Ukraine, showed continuity over 4000 years, from the Mesolithic to the end of the Neolithic, in contrast to almost all other parts of Europe where population turnover drove this cultural change, including vast areas of Central Europe and around the Danube River.

[1] Human Evolution, Department of Organismal Biology, Uppsala University, 75105 Uppsala, Sweden. [2] Institute of Human Biology & Evolution, Faculty of Biology, Adam Mickiewicz University in Poznań, 61-614 Poznań, Poland. [3] Department of Archaeology and Ancient History, Uppsala University, 75126 Uppsala, Sweden. [4] Geography Research Unit, University of Oulu, 90014 Oulu, Finland. [5] Kerttu Saalasti Institute, University of Oulu, 90014 Oulu, Finland. [6] Department of Anthropology and Archaeology, University of Bristol, Bristol, UK. [7] Institute of Archaeology, University of Gdańsk, 80-851 Gdańsk, Poland. [8] "Francisc I. Rainer" Institute of Anthropology, Romanian Academy, 050711 Bucharest, Romania. [9] Faculty of History, University of Bucharest, 030167 Bucharest, Romania. [10] National History Museum of Transylvania, Cluj-Napoca, Romania. [11] Vasile Pârvan Institute of Archaeology, Bucharest, Romania. [12] Institute of Archaeology, Jagiellonian University, 31-007 Kraków, Poland. [13] Karkonosze Museum, 58-500 Jelenia Góra, Poland. [14] Polish Academy of Sciences, Institute of Archaeology and Ethnology, 31-016 Kraków, Poland. [15] Museum of Folk Culture, 11-600 Węgorzewo, Poland. [16] Faculty of Archaeology, Adam Mickiewicz University in Poznań, 61-614 Poznań, Poland. [17] Department of Bioarchaeology, Institute of Archaeology, National Academy of Sciences of Ukraine, 04210 Kyiv, Ukraine. [18] Department of Physical Anthropology, Institute of Forensic Medicine, University of Bern, 3008 Bern, Switzerland. [19] Archaeological Museum, 61-781 Poznań, Poland. [20] Institute of Biological Sciences, Cardinal Stefan Wyszyński University in Warsaw, 01-938 Warszawa, Poland. [21] Centre for Palaeogenetics, Stockholm University and the Swedish Museum of Natural History, 106 91 Stockholm, Sweden. [22] Department of Archaeology and Classical Studies, Stockholm University, 106 91 Stockholm, Sweden. [23] Department of Internal Medicine and Radboud Center for Infectious Diseases, Radboud University Medical Center, 6525 HP Nijmegen, the Netherlands. [24] Department for Genomics & Immunoregulation, Life and Medical Sciences Institute (LIMES), University of Bonn, 53115 Bonn, Germany. [25] Grand Valley State University, Department of Biology, Allendale, MI 49401, USA. [26] Museum of Cultural History, University of Oslo, 0130 Oslo, Norway. [27] Centre for Anthropological Research, University of Johannesburg, Auckland Park, 2006 Johannesburg, South Africa. [28] SciLifeLab, Uppsala University, 75105 Uppsala, Sweden. ✉email: tiina.maria.mattila@ebc.uu.se; mattias.jakobsson@ebc.uu.se

Modern humans started spreading into Europe some 50,000–40,000 years ago[1–3]. Before the agricultural transition that started ~8500 years ago[4,5], Europe was inhabited by hunter-gatherer populations, roughly clustering into two groups (as defined by archaeogenetics); Western Hunter-Gatherers (WHG) in Western Europe and East European Hunter-Gatherers (EHG)[6–8] in northeastern and in the extreme eastern frontier of Europe[9,10]. In between these core regions, the groups from the east (EHG) and from the west (WHG) probably met and admixed[11–13]. In Scandinavia, where ice coverage partially persisted until 10,000 years ago, the colonization of WHG groups took place from the south, whereas EHG groups entered from the northeast, likely following the Norwegian Atlantic coast from the north to the south, creating an admixture pattern that goes in the opposite direction to central/eastern Europe[11]. However, our knowledge concerning the history and dynamics as well as the time scale of genetic admixture and continuity of the Mesolithic populations across Europe is still limited.

The population structure of Stone Age Europe experienced a large-scale change in the early Holocene. This change was driven by the migration of farming groups (European Neolithic, EN)[14–16] from the Near East, which were genetically closely related to the groups from the Neolithic Anatolia (AN)[17–19] and more distantly to the hunter-gatherers from the Caucasus region also known as CHG[20]. The mode and level of population interaction in the initial and subsequent times of the European Neolithic farmers and hunter-gatherers has been a matter of debate for very long time. The current consensus points to geographically and temporally varying level of genetic admixture of the EN and WHG groups[7,21–24] starting already at the early stages of the arrival of the former in central Europe[24]. Based on evidence from the archeological record, there may have been differences in the levels of cultural contacts between the farmer and hunter-gatherer groups in a west-east gradient of the widely spread Early Neolithic Central European Linear Pottery culture (LBK)[25]. However, the suggested differences in interaction levels may have been in the form of exchange of goods rather than genetic admixture.

In addition to the variable contacts and interactions between the hunter-gather and incoming farmer groups, in some European regions (for instance in parts of Scandinavia, the Baltic region, and the Eastern Europe) the hunter-gathering lifeway prevailed for much longer in comparison with the Southern and Western Europe. In Ukraine for example, the steppe and forest steppe zones of the North Pontic region were inhabited by hunter-gatherer communities still during the Neolithic sustaining mostly on aquatic resources[26]. A similar type of development took place in these communities as in the Neolithic farming groups. For instance, in some parts of Eastern and Northeastern Europe pottery was introduced but the subsistence was still mainly based on hunting and gathering[27,28]. Genetic data from some of these groups have shown that the genetic makeup before and after the European agricultural dawn remained similar in contrast to Central and Western Europe[12,29,30].

To improve our understanding of the level, character, and regional variability of contacts between the Central and Eastern European Stone Age groups, we sequenced and analyzed whole genomes of individuals who lived before and after the Neolithic transition (i.e., 7500–5500 cal BP) in the eastern frontier of Europe. The investigated area encompasses an area covering modern-day Romania, Poland, and the lower Dnipro Valley region in Ukraine over a time span of ~5000 years (ca. 10,500–5500 cal BP). Our investigation revealed that before Neolithic, the eastern frontier of Europe contained an admixture cline between genetically differentiated groups from Central Europe and Siberia. We also observe stronger genetic continuity and limited admixture in the Dnipro Valley region after the Neolithic transition while large-scale gene flow took place in populations further to the west.

## Results and discussion

To investigate the genetic affinities in Stone Age Central and East Europeans, we generated genome-wide sequencing data from a collection of 56 individuals from Epipaleolithic/Mesolithic, Neolithic, and Eneolithic Poland, Romania, and Ukraine (Fig. 1a, b; see Supplementary Note 1–4, Supplementary Figs. 1–11, and Supplementary Data 1 and 2). The depth of coverage per individual ranged from 0.01 to 4.55X (Supplementary Data 1).

**Over 4000 years of genetic continuity in the Stone Age lower Dnipro Valley region in Ukraine**. To characterize the genetic structure of our data, we first used a principal component analysis (PCA) for the dataset together with a collection of Stone Age and Bronze Age individuals across West Eurasia (Supplementary Data 3). The PCA placed all the Epipaleolithic/Mesolithic Central and East European individuals on a cline between WHGs (represented by individuals Bichon, Loschbour, Ranchot88, Rochedane, and Villabruna[6,8,20]) and the Upper Paleolithic Siberian Afontova Gora3[8] (Fig. 2a), consistent with previous findings[10,12]. Comparative Mesolithic hunter-gatherers from Western Russia (EHG and WRuHG), the Baltic region (BHG), and Sweden & Norway (SHG) also fell within this cline. To gain further insight into the genetic composition of the studied groups, we inferred ancestry components[31] (Fig. 2b), including a broader set of comparative individuals from the Stone Age and modern times, sampled across Eurasia (see "Material and methods" section and Supplementary Data 3 for details). The individuals from the Neolithic lower Dnipro Valley were genetically very similar to the Epipaleolithic/Mesolithic individuals from this region. In contrast, the Neolithic/Eneolithic individuals from the Romanian and Polish sites displayed the same ancestry components as other European farming groups, and were genetically similar to the Anatolian Neolithic farmers[17–19] (Fig. 2a, b). These results were also supported by the patterns of allele sharing with WHG and EN (see Fig. 3a–c where positive values indicate closer affinity to WHG and negative values to EN) as well as the uniparental markers (Supplementary Note 5 and Supplementary Data 1, 4, and 5).

To test for genetic continuity in the three regions investigated in this study, we calculated the level of shared genetic drift with the local Mesolithic individual through time using the $f_3$-outgroup test $f_3$(Yoruba; X, Y), where X was the test individual and Y the highest genome coverage Mesolithic individual from the same region. In addition, we employed the co-called Anchor Method to assess population continuity through time[32] using Loschbour (WHG) as an anchor individual. These tests further supported strong genetic continuity of the lower Dnipro Valley region from the Mesolithic to the Neolithic (Fig. 3f and Supplementary Fig. 12). The difference between the oldest dated and the youngest dated individuals (ukr125: 10,547 cal BP, ukr123: 6233 cal BP; Supplementary Data 1) showed that the genetic continuity in this region lasted more than 4000 years. In contrast, for the Romanian and Polish individuals, there is a distinct genetic discontinuity between the pre-Neolithic and the Neolithic individuals, caused by population turnover in these regions (Fig. 3d, e and Supplementary Fig. 12).

From the genome sequence data, we can also assess genetic diversity (conditional nucleotide diversity, CND[33]), which indicates past effective population sizes. To get an indication of past population sizes, we compared CND trough time in each region. The magnitude of CND was very similar for the Mesolithic and Neolithic populations from the Dnipro Valley region, in contrast to Romania and Poland where the genetic diversity is much higher among the Neolithic individuals in comparison with any Mesolithic

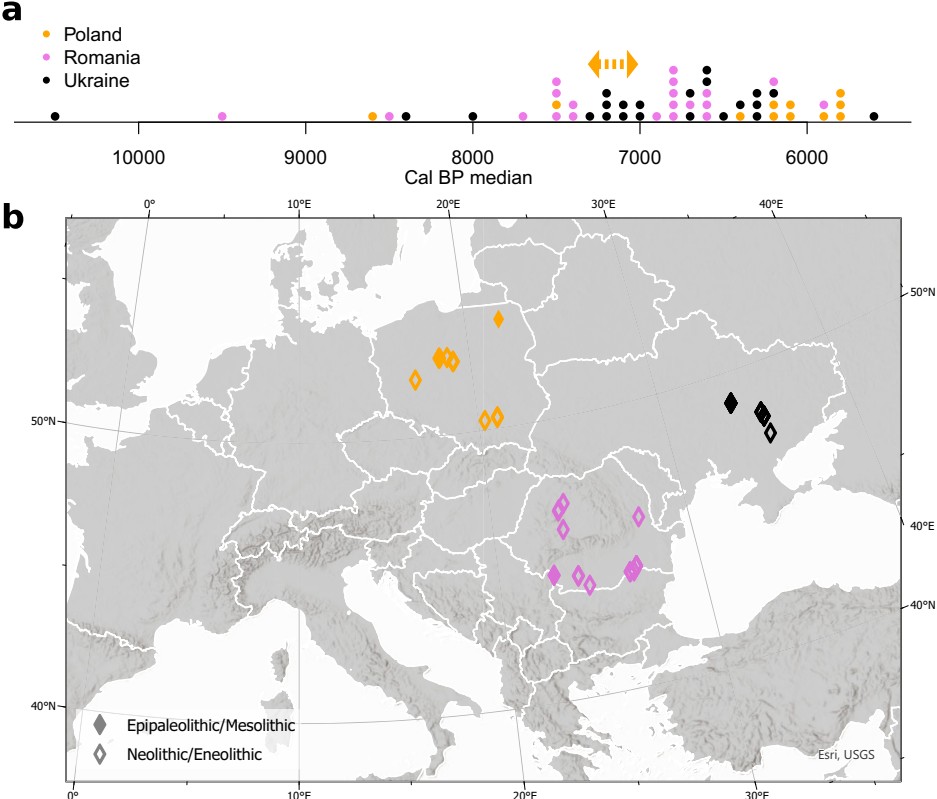

**Fig. 1 Summary of the individuals sequenced in this study. a** The distribution of cal BP median of 53 dated individuals. The orange arrow shows the context-based approximate age of three LBK samples (lbk101, lbk102, and lbk104). **b** Geographic location of the newly sequenced individuals. The map graphic was produced using ArcGIS® pro software by Esri. ArcGIS® pro and ArcMap™ are licensed through Esri. Copyright © Esri. All rights reserved. See www.esri.com for more information about the Esri® software. The basemap was obtained from World Bank Official Boundaries collection[99].

estimate (Fig. 4a). Hence, we concluded that the Dnipro Valley population likely stayed relatively stable in size (at least in terms of effective population size) and was unaffected by admixture with European Neolithic farmers/Anatolian farmers.

**Isolation-by-distance in Mesolithic Western Eurasia.** To verify the potential admixture between the Upper Paleolithic Siberian group and WHG, we first tested if Loschbour (representing WHG) forms a clade with the European Mesolithic individuals in respect to the AfontovaGora3 (representing the Upper Paleolithic Siberian group) using $f_4$-test (chimp, AfontovaGora3; X, Loschbour) and the Human Origins overlap panel. The $f_4$-values were negative for all test individuals indicating increased allele sharing with the Paleolithic Siberian AfontovaGora3, indicating gene flow between the WHG and Paleo-Siberian lineage. However, the tests were not significantly different from 0 for the Polish Mesolithic individuals, indicating genetic similarity to WHGs (Supplementary Fig. 13a). To increase the power of the test, we calculated $f_4$(chimp, Sidelkino; X, Loschbour) from the 1000 genomes overlap panel and confirmed the significant contribution from the eastern lineage to all Central and East as well as North European Mesolithic individuals investigated (Supplementary Fig. 13b). In the latter case the Mesolithic Sidelkino[9] represents the EHG and Loschbour the WHG. A model-based two-source analysis separated the admixture model (WHG-AfontovaGora3) from the best single-source models in 19 cases (nested $p$-value < 0.05) and in 15 cases both single-source models were rejected (tail probability < 0.05). The estimated admixture proportions of WHG-related ancestry ranged from 50.9% (40.9–60.9%, 95% Jackknife CI) for Sidelkino to 88% (76.2–99.8%) for SC1 (Supplementary Data 6).

The different admixture models between the Paleo Siberian-WHG gradient were also tested (using qpGraph[34]) including representative groups from the gradient. The stepping stone-like graph (Fig. 4b) including admixture from a group related to the Paleolithic Siberian (represented by AfontovaGora3) in EHG (represented by Sidelkino) and this lineage further re-admixing with the WHG lineage was consistent with the data (worst Z-score 0.978, $f_4$(Sidelkino, Loschbour; ukr102, ble008)). Furthermore, as three other tested models without this admixture were inconsistent with the data (Supplementary Fig. 14), the admixture between the West European and Siberian lineages was further strengthened. The connection between the EHG and the Paleolithic Siberian lineage has been reported also in Fu et al.[8], but it was not clear that EHG is part of the Paleo Siberian-WHG gradient previously.

The patterns of genetic admixture in the Mesolithic of the European continent suggest a geographical dependency in the Paleolithic Siberian-WHG ancestry proportions. Previous archaeogenetic analysis has indicated that the Eastern and Western Hunter-Gatherer lineages were admixed in Scandinavia forming an EHG/WHG gradient in Northern Europe[11]. We tested the fit of the isolation-by-distance admixture model (admixture IBD) in the Paleo Siberian-WHG cline using a linear regression analysis of the level of allele sharing ($f_4$-test) and distance from the approximate region occupied by unadmixed WHG population. The geographic coordinates of a set of samples assigned as WHG[8] were used to approximate this so-called WHG core region (Supplementary Fig. 15). As a measure of the distance from the WHG core region, we took the shortest optimal topology aware route from five WHG points (see Methods and Supplementary Fig. 15 for details). The linear regression analysis indicated a significantly decreasing

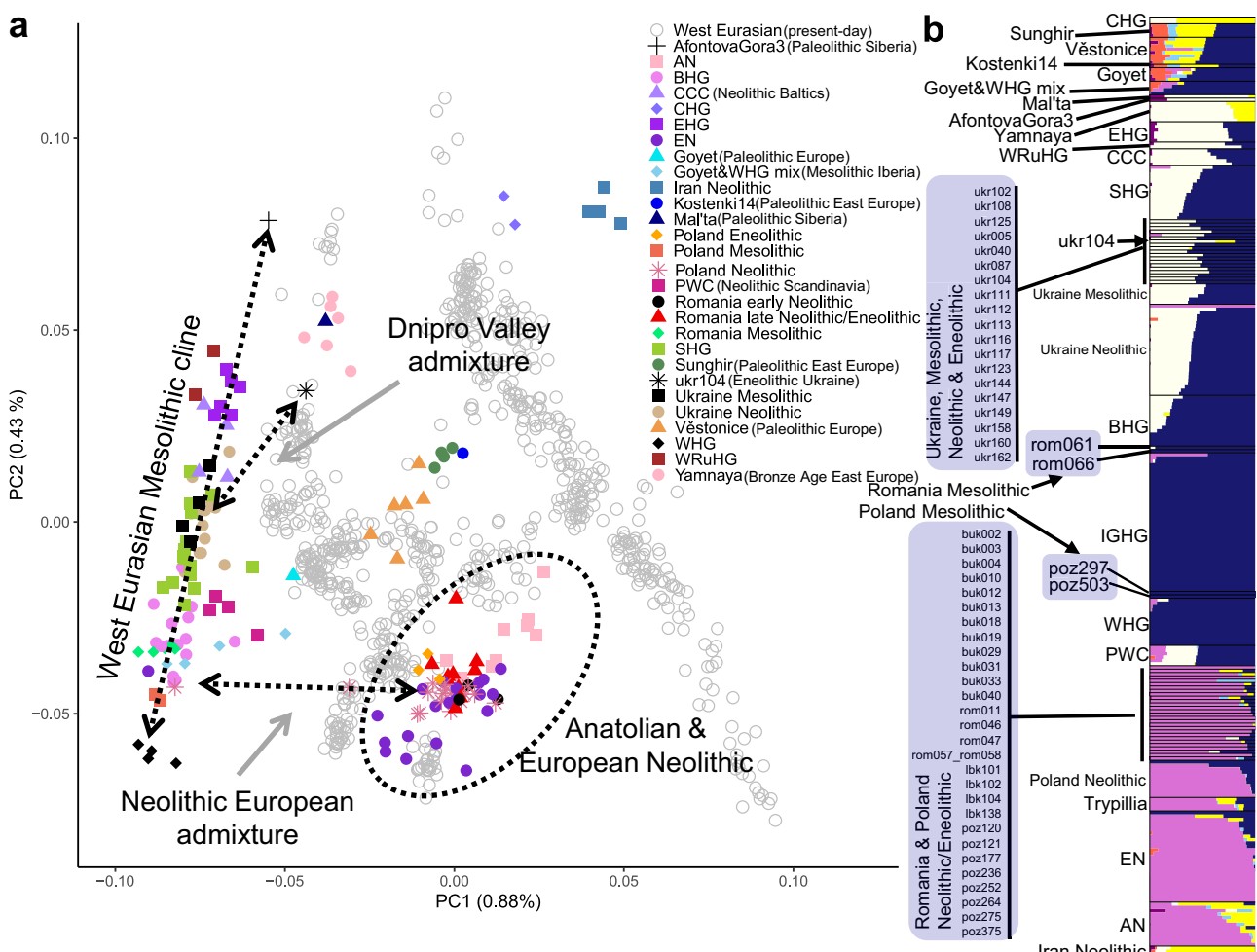

**Fig. 2 Population structure of Stone Age West Eurasians. a** Principal component bi-plot of selected Paleolithic, Mesolithic, and Neolithic West Eurasian individuals projected onto eigenvector space estimated from a set of modern-day West Eurasian groups from the Human Origins dataset[6]. Only individuals who have at least 10,000 called SNPs are shown on the plot. Arrows and the black circle highlight focal groups and individuals investigated in this study. Full ancient individual annotation is available from Supplementary Fig. 19. **b** Estimated ancestry proportions for the ancient individuals assuming seven ancestry components. The lab IDs are shown for individuals from this study (shaded). The full Admixture plot is available from Supplementary Fig. 20. AN anatolian neolithic, BHG Baltic hunter-gatherers, CCC comb ceramic culture (neolithic hunter-gatherers) from the Baltics, CHG Caucasus Hunter-Gatherers, EHG Eastern Hunter-Gatherers, EN European Neolithic, IGHG Iron Gates Hunter-Gatherers, PWC Pitted Ware Culture (Neolithic Hunter-Gatherers) from the Scandinavia, SHG Scandinavian Mesolithic Hunter-Gatherers, WHG Western Hunter-Gatherers, WRuHG West Russian Hunter-Gatherers.

proportion of the WHG ancestry and increasing Paleolithic Siberian (represented by AfontovaGora3) ancestry in West Eurasia as a function of minimum distance from the WHG core region (linear regression coefficient for minimum distance $= -9.0 \times 10^{-6}$, SE $= 1.7 \times 10^{-6}$, $t$-value $= -5.3$, $p$-value $= 2.5 \times 10^{-5}$; Fig. 4c and Supplementary Fig. 16). The results were significant also after removing the possible leverage points from the analysis (linear regression coefficient for minimum distance $= -6.5 \times 10^{-6}$, SE $= 3.0 \times 10^{-6}$, $t$-value $= -2.1$, $p$-value $= 0.048$; Supplementary Figs. 17 and 18). Similar results were obtained if the qpAdm estimated WHG ancestry proportions (instead of $f_4$) or the WHG distance median or total distance from all the WHG points (instead of minimum distance form WHG) were used in the modeling (Supplementary Data 7).

Gene flow between two genetically differentiated populations is also expected to increase genetic diversity as previously observed in Scandinavia[11]. The highest diversity is expected when the ancestry proportions are close to equal given other population processes being equal. This was evaluated by comparing the conditional nucleotide diversity estimates in Mesolithic groups

with different admixture proportions. In line with the expectation, we observed a decrease in diversity as a function of the level of EHG admixture (Fig. 4a). Taken together, the expectations of the IBD admixture model indicate long-distance, stepping-stone-like gene-flow between Europe and Siberia in pre-Neolithic Europe.

**Gene flow to the lower Dnipro Valley population.** Even though the major ancestry components of the Mesolithic and Neolithic lower Dnipro Valley population derived from WHG and Paleolithic Siberian lineages (where EHG likely functioned as a stepping stone), we also found that a three-way population admixture model that includes the Caucasus Hunter-Gatherers (EHG-WHG-CHG) fits the genetic ancestry composition of this population (Supplementary Data 8). We estimated that ~7.4% (0.15–14.7%, Jackknife 95% CI) of the genetic ancestry in the Dnipro Valley population is derived from a CHG population indicating a genetic connection between the Caucasus and the North Pontic region in the Mesolithic/Neolithic. The allele

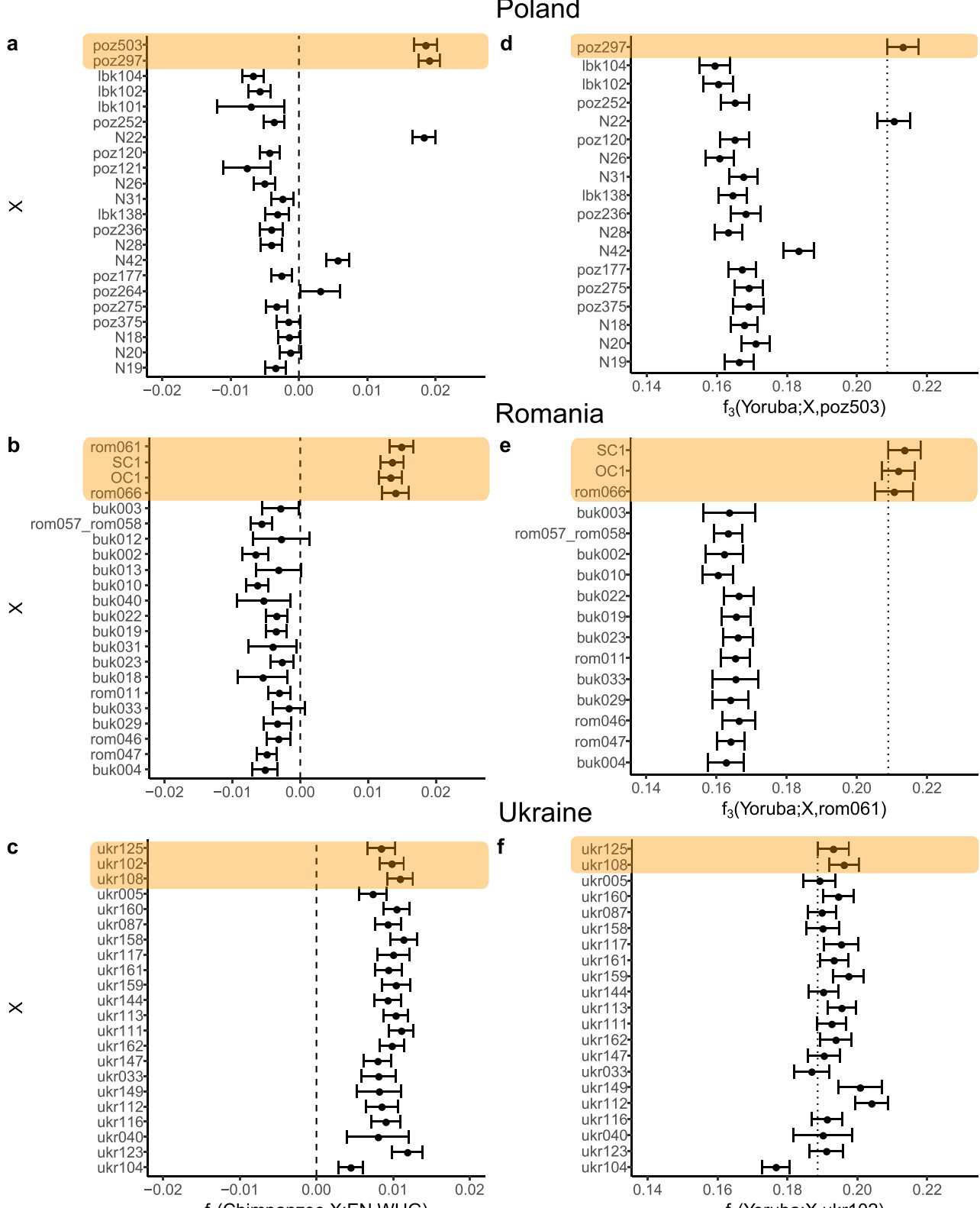

**Fig. 3 Patterns of allele sharing in Mesolithic, Neolithic, and Eneolithic Central and Eastern Europeans.** (**a**-**c**) $f_4$-statistics testing allele sharing between Mesolithic Central European hunter-gatherers (WHG, Loschbour) and European Neolithic farmers (EN, LBK) (**d**-**f**) Regional continuity $f_3$-outgroup test. The vertical line shows the lower point of the 95% confidence interval for the comparison with the oldest dated individual. The individuals included were excavated in modern-day Poland (**a** & **d**), Romania (**b** & **e**), and Ukraine (**c** & **f**). The data are shown for newly produced data and, additionally, for three Mesolithic Romanian (sample labels OC & SC) and eight Neolithic individuals from Poland (sample label N) published previously[22,36]. Error bars indicate the 95% confidence intervals from block Jackknife standard errors. The individuals were ordered based on their cal $^{14}$C age 95,4% range midpoint or context-based age midpoint (lbk101, lbk102, lbk104). All the statistics were calculated using the 1000 genomes transversion overlap panel. Only tests which are based on at least 10 000 (for $f_4$) and 500 (for $f_3$) sites are shown. Patterns of allele sharing in Mesolithic are highlighted in orange.

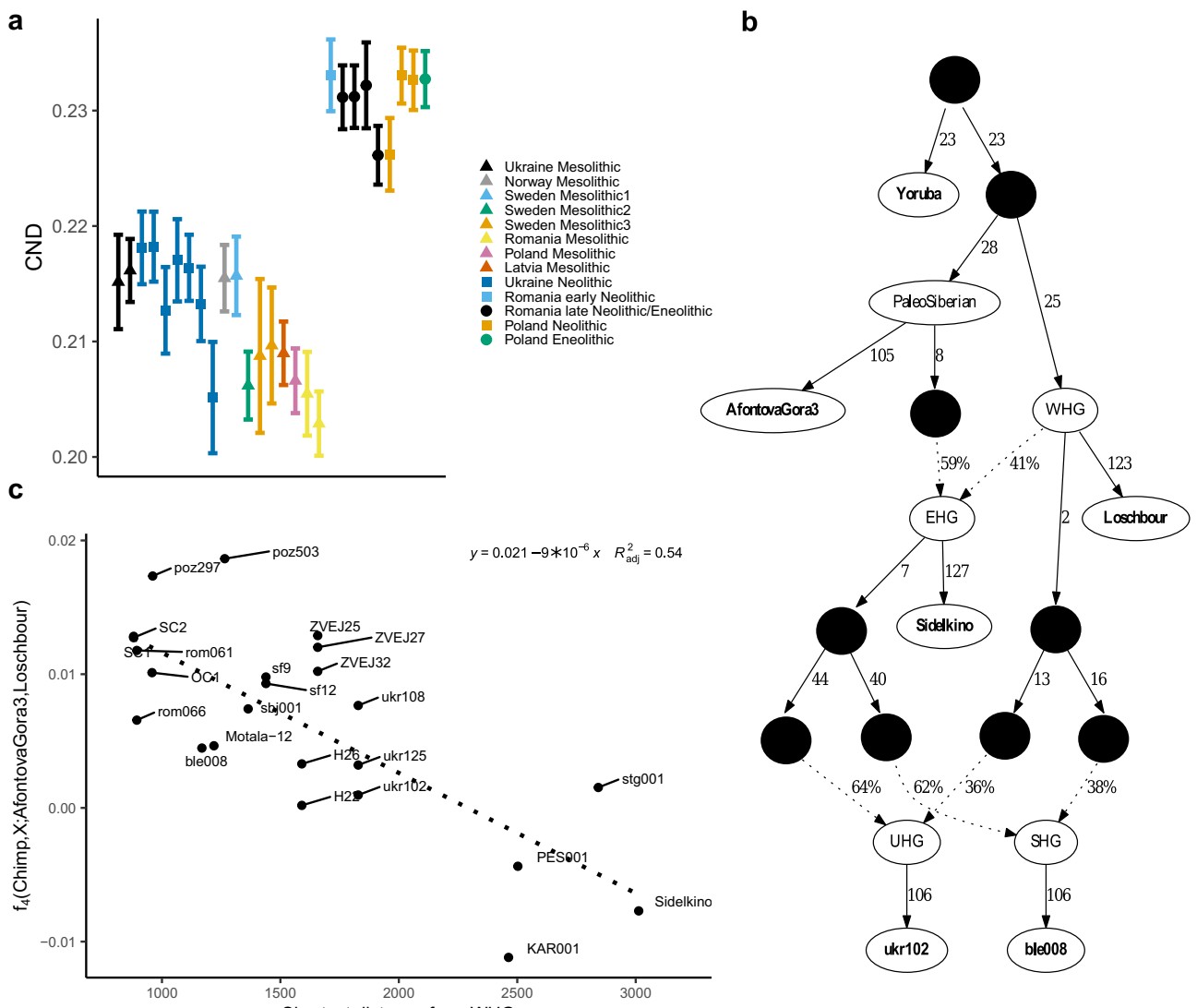

**Fig. 4 Admixture and genetic diversity in Stone Age Europeans. a** Conditional nucleotide diversity for selected Mesolithic, Neolithic and Eneolithic European individuals. The individual pairs included in this analysis are available at Supplementary Data 14. The Mesolithic individuals from the region of Sweden were split into three groups Sweden Mesolithic1: two individuals from Huseby Klev[80]; Sweden Mesolithic2: two individuals from Gotland[11]; Sweden Mesolithic3: four individuals from Motala[6]. The error bars represent 95% confidence intervals from standard errors. **b** qpGraph model including stepwise admixture between Paleo Siberian (represented by AfontovaGora3), Western Hunter-Gatherers (WHG, represented by Loschbour), and Eastern Hunter-Gatherers (EHG, represented by Sidelkino). The additional admixture nodes included here were the Ukrainian Mesolithic (UHG, represented by ukr102) and Scandinavian Mesolithic (SHG, represented by ble008) individuals. The data-point nodes are in bold. **c** Scatterplot and linear regression model of distance from the closest WHG data-point and allele sharing ($f_4$) between WHG (Loschbour) and Paleo Siberians (AfontovaGora3).

sharing with CHG was significantly higher among the Neolithic Dnipro Valley individuals (Supplementary Data 9) which means that at least some level of this ancestry sharing is due to mixing during the Neolithic.

In addition, the Eneolithic individual from the lower Dnipro Valley region (Deriivka II cemetery) archeologically classified as Serednyostogivs'ka (Sredny Stog) horse keepers (ukr104, c. 5650-5477 cal BP) showed smaller level of allele sharing with other individuals from the same region (Fig. 3f). This indicates gene flow from a population that is genetically differentiated from the preceding local population. This individual (ukr104) was genetically more similar to the Bronze Age Yamnaya individuals from Samara, the CHG, and the Neolithic Iranian than the other Dnipro Valley samples (Fig. 2a, b). To test this possible gene-flow, we modeled ukr104 as a mixture of a set of lower Dnipro Valley individuals (ukr087, ukr102, ukr111, ukr113, ukr160) and

Yamnaya[35] using qpAdm[34]. Other ancient neighboring groups AN, CHG, EHG, Neolithic Iranian WC1, Mal'ta, WHG, and Sunghir were used as reference ('right') populations in addition to a chimpanzee outgroup (Supplementary Data 10). The admixture model fitted the data ($\chi^2 = 2.37$, tail probability = 0.88, df = 6), while the single-source models were rejected (tail probability < 0.05, Supplementary Data 10). The estimated admixture proportions were 33.2% (25.0–41.4%, 95% Jackknife CI) of the local Meso-Neolithic Dnipro Valley ancestry and 66.8% (58.6–75.0%) of the Yamnaya related ancestry.

**Admixture through time in the Neolithic Central and Eastern Europe**. To explore the admixture between the Neolithic East European and the descendants of European Mesolithic hunter-gatherer groups, we tested if the hunter-gatherers from Poland and

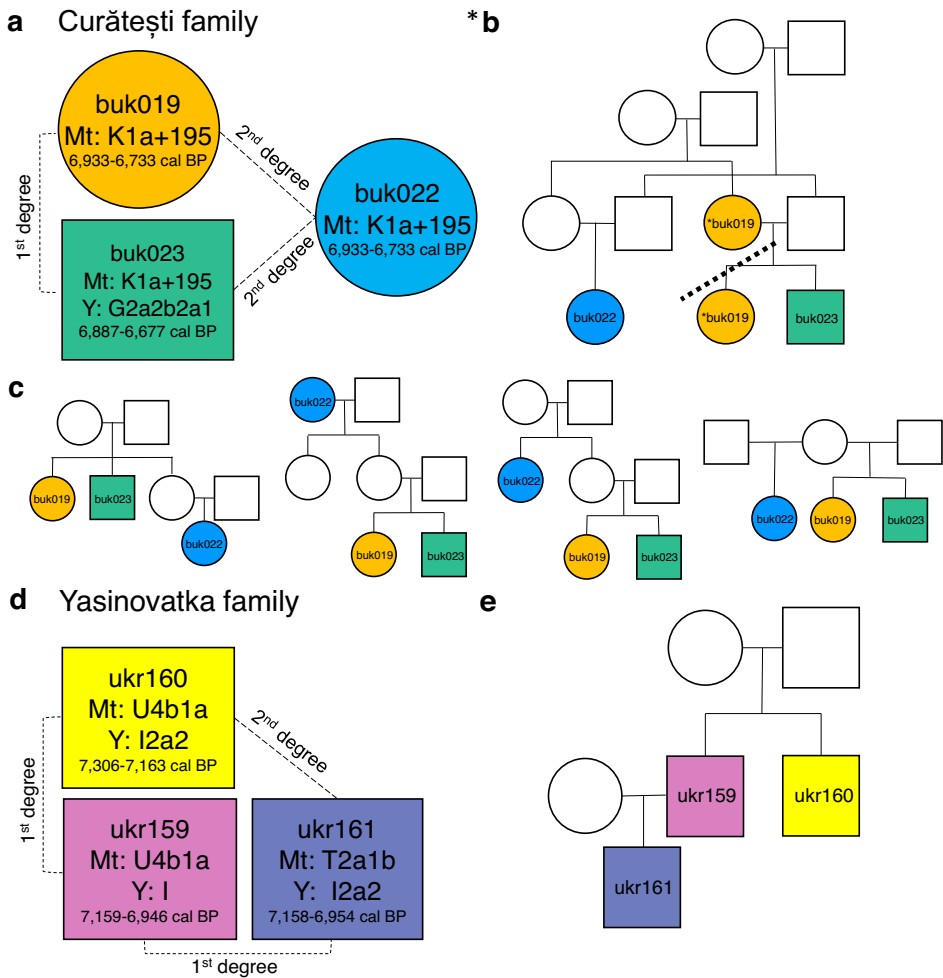

**Fig. 5 Family trios from Neolithic Ukraine and Romania. a** Information on the Curătești family members. **b, c** Six possible genealogical models of the Curătești family. *Two very similar double cousin scenarios where buk019 is either the mother or sister of buk023 are depicted on the same figure (**b**) where the dashed line separates the two alternatives. **d** Information on the Yasinovatka family members. **e** Suggested genealogy of the Yasinovatka family.

Romania (poz297 and rom061, respectively) share more alleles with the Romanian and Polish Neolithic/Eneolithic individuals when compared with early Neolithic Central Europeans. In comparison with the early Neolithic LBK individual from Germany[6], a significant increase in allele sharing with the local hunter-gatherers were detected in 16 out of 30 investigated Neolithic/Eneolithic individuals from Poland and Romania (Supplementary Data 11).

The estimated ancestry deriving from the local Mesolithic hunter-gatherers (Z-score > 2, $f_4$-ratio test) ranged from 9 to 20% in the Romanian Neolithic/Eneolithic individuals while it was 9–97% among the Neolithic/Eneolithic individuals from Poland (Supplementary Data 11). We also observed a significant increase in the proportion of admixture through time (linear regression coefficient for $^{14}$C midpoint = $-4.7 \times 10^{-5}$, SE = $1.8 \times 10^{-5}$, t-value = $-2.7$, p-value = 0.013; N22, N42, and poz264 excluded with the most extreme α values). This resurgence of the local Mesolithic ancestry in the Eneolithic has also been found in previous studies in other parts of Europe[23,36].

**Kinship in Stone Age Europe.** The patterns of genetic kinship in pre-historic societies can inform us about their social organization. Hence, we also investigated close kinship (1st and 2nd degree kin relations) among the studied individuals within population using the READ software package[37]. We detected two kin trios (standard error scaled distance normalized mean P0 score > 1.96) among the sequenced individuals (Supplementary Data 11). The first trio from

the Boian context from Curătești (Romania) included two adult females and one adult male (buk019, buk022, buk023, hereafter Curătești family). The second trio of adult males was found among the individuals from Yasinovatka, Ukraine (ukr159, ukr160, ukr161, hereafter Yasinovatka family; Fig. 5a, d). All data for the detected kin were derived from single bone specimens and single extracts for each individual.

From the Curătești family, buk019 and buk023 were first-degree relatives, while buk019 and buk023 were second-degree relatives to buk022 (Supplementary Data 12). All three carried mt haplogroup K1a + 195 (Supplementary Data 1 and 4) suggesting that they were maternally related (Fig. 5a–c). Assuming that the shared uniparental haplogroups indicated direct matri- and patrilineality, we constructed possible genealogies for the detected families. The kinship assignments are consistent with the genealogical models where buk022 was a grandmother or an aunt of the siblings buk019 and buk023 from their mother's side. Equally possible models are that buk022 was a niece of the siblings buk019 and buk023 from their sister's side, buk022 was a maternal half-sib of the full-sibs buk019 and buk023, or buk22 was a double-cousin of the full-sibs buk019 and buk023 or, alternatively, buk022 was a double-cousin of buk023 and niece of buk019, and buk023 was a mother of buk019 (Fig. 5b, c). The radiocarbon inferred age estimates overlapped for all three individuals (Supplementary Data 1).

The Yasinovatka family members were all adult males (Supplementary Data 1 and Supplementary Note 3). We found that ukr160

and ukr161 were second-degree relatives, while ukr159 was a first-degree relative of both ukr160 and ukr161 (Supplementary Data 12). Two individuals in this trio (ukr159 and ukr160) had U4b1a mt haplogroup, and the third had T2a1b (Fig. 5a and Supplementary Data 1 and 4) indicating a non-maternal relationship between the first two and ukr161. The Yasinovatka family members' Y-haplogroups fell within the I clade (Fig. 5d and Supplementary Data 1 and 5), suggesting a possible patrilineal relationship. The difference in Y-haplogroup assignment precision likely explains the difference in the final haplogroup assignments (Fig. 5d) since no data were available on the I2a2 defining mutations for the low coverage ukr159 (Supplementary Data 5). Despite the occasional difference in the called Y genotypes, we concluded that I was the most likely Y haplogroup for all of the Yasinovatka family members (Supplementary Data 5 and 13). These results are compatible with a model where ukr159 and uk160 were brothers, and ukr161 was the son of ukr159 (Fig. 5e). The two brothers were buried in adjacent pits associated with the earliest stage at the site, and, based on the [14]C date (Fig. 5d), ukr160 likely died slightly earlier than ukr159. While ukr160 was buried alone, the brother ukr159, was buried in a pit together with his son (ukr161) and at least two other individuals not analyzed in this study[38]. Two additional males, both unrelated to the Yasinovatka family and representing different paternal lineages, were analyzed in this study (Supplementary Data 1). One of them, ukr158, was also associated with the earliest stage at the site and buried in a pit adjacent to ukr159/ukr161 together with five other individuals. The other, ukr162, was one of thirty burials in a larger pit that belonged to a later stage of interments. An additional kin pair was detected among the previously published Stone Age Ukrainian Dnipro Valley individuals[12] and the dataset from this study (Supplementary Data 1). This pair was a first-degree kin from Mesolithic Deriivka I (ukr102 from this study & I5876 from Mathieson et al.[12]). Both analyzed individuals were males who carried the same mt & Y haplogroups (Supplementary Data 1, 4, and 5). These findings are in line with the genealogy where these two individuals were brothers. They were both of similar age (between 40–50 years old) and found in a joint burial pit together with a younger individual not analyzed here[39].

Among the individuals from Poland, we did not find any first- or second-degree kin pairs (Supplementary Data 12). Interestingly, two of the samples (poz120 & poz121, see also Supplementary Note 1) from the Krusza Zamkowa 3 cemetery were buried in close proximity, which has earlier been suggested to indicate their biological relatedness[40]. Similar to the results from Juras et al.[41], we can conclude that these two females were not genetically related, at least not in the form of full siblings, mother-daughter, aunt-niece or grandmother-granddaughter. These individuals were also not related to the adult female lbk138 buried ~25 meters away. These burials are exceptionally richly equipped, with similar types of beads and adornments and have even been designated as so-called princess graves[42]. A lack of maternal kinship among individuals buried close to each other have previously been found among LBK in Karsdorf in Germany[43]. Thus, social ties rather than genetic kinship - may have been of importance in burial arrangements in the Krusza Zamkowa community[44]. Different non-biological relations among individuals in pre-historic burials have recently been discussed[45]. It has also been hypothesized that other factors, related to socioeconomic organization possibly linked to specific activities, may have played a role for burial practices[46,47].

## Conclusions

In this study, we have investigated the genetic landscape of Central and Eastern Europe before and after the European Neolithic expansion. One of the most striking findings was that before the dawn of the European Neolithic, Central and Eastern Europe was inhabited by a population that descends from a gradient admixture population between genetically distinct West European and Siberian hunter-gatherer groups. Such a pattern suggests long distance population genetic connectivity, likely via a stepping-stone admixture model. The genetic descendants of these Mesolithic populations were in many areas assimilated or replaced by incoming farmers during the Neolithic, and the genetic group common during the late Mesolithic remained dominant only in the East and Northeast European frontier and some geographical regions in Southern Scandinavia. In the lower Dnipro Valley region in Ukraine, the direct descendants of the Mesolithic population continued being the dominant group for thousands of years after the start of the European Neolithization, and the end of this continuity was associated with the Eneolithic/Bronze Age migration wave from the East. Hence, we conclude that the Dnipro Valley region's Neolithic cultural innovations, such as adoption of pottery (further from pointed-bottom vessels to flat bottomed ones), pioneer animal husbandry (cattle, pig, sheep & goat, agriculture e.g., barley)[48] and the changes from contracted to extended supine burials were not associated with gene flow from Anatolia. This is opposite to the pattern observed in regions further to the west, where Neolithic transition was associated with large-scale migration.

Our analysis of close genetic relatedness, on the one hand, revealed the role of genetic relatedness in burial practices in cultures across Mesolithic, Neolithic, and Eneolithic Europe. One the other hand, the results also pointed to a possibility of non-genetic connections such as in the Neolithic Late Lengyel culture Kruza Zamkowa case exemplified here. These observations, together with previous investigations of close kin relations in the Stone Age[49–52], suggest a variety of different views and practices of biological and potentially non-biological kin relations.

## Methods

**Ethical statement**. The study has conducted following the 'Ethical guidelines for good archaeological praxis', including respectful handling of archaeological human remains, as described by the Swedish Archaeological Society[53]. The sampling appropriateness of the material was overseen by the museum curators included in this study.

**Sampling and radiocarbon dating**. A set of 59 prehistoric human bone specimens (of which 56 were confidentially assigned as independent individuals) that have been attributed to either the Mesolithic (/Epipaleolithic), the Neolithic or the Eneolithic periods, was collected from Poland, Romania, and Southeastern Ukraine (Supplementary Data 1 and 2). To assess the age of each sample, [14]C was measured from tooth or bone specimens in Beta Analytic Carbon Dating Service in Miami, FL, USA (44 specimens) and in Poznań Radiocarbon Laboratory in Poland (7 specimens) or collected from the literature (3 specimens). For a subset of samples (see Supplementary Data 1), carbon ($\delta^{13}$C) and nitrogen ($\delta^{15}$N) isotope ratios were measured using Isotopic Ratio Mass Spectrometry (IRMS). This was carried out at Beta Analytic Carbon Dating Service for most of the samples and the remining at the Department of Geosciences at the Faculty of Geosciences/Geography at the Goethe University in Frankfurt am Main, Germany. The dates were calibrated using Oxcal v4.4.4, IntCal 20 and freshwater reservoir effect (FRE) correction was applied sample specifically depending on the stable isotope-based dietary analysis. See Supplementary Notes 1–4 and Supplementary Data 15–17 for detailed information on the collagen quality control, diet inference, FRE correction, and the archeological background of the samples.

**Generating aDNA sequence data**. All pre-PCR laboratory work was conducted in dedicated ancient DNA laboratories, with UV-lamps in the ceiling (254 nm), positive air pressure, and HEPA-filtered laminar flow hoods, either at Uppsala University, Sweden (buk-, rom-, and ukr-specimens) or at Adam Mickiewicz University in Poznan, Poland (lbk- and poz-specimens). The laboratories were frequently cleaned with bleach (NaOCl) and UV-irradiation. Equipment and non-biological reagents were regularly decontaminated using bleach and/or RNase AWAY™ (ThermoScientific) and UV irradiation.

Prior to sampling, bones and teeth were decontaminated through UV-irradiation (254 nm, 6 J/cm[2] on each side), wiping with 1 % bleach, a second round of UV-irradiation and finally by removal of the outer surface layer. Two strategies were used when sampling for DNA. Either we removed between 40–110 mg of bone- or tooth root powder (with a Dremel drill or by grinding down a bone piece

with a Starbeater mill from VWR), or we removed a small amount of bone or tooth root of approximately the same weight using a Dremel drill with diamond cutting wheels. DNA was extracted using silica-spin column protocols, either following protocol C[54] but with the SDS in the extraction buffer exchanged for 1 M Urea as in[55] or by using a protocol targeting shorter fragments[56] and finally eluted in 40–110 µl EB (Qiagen). When DNA was extracted from bone fragments, the protocols were modified to include an initial incubation step with 1 mL 0.5 M EDTA in a hybridization oven at 37 °C and on rotation for 30 min, followed by removing the solution. If the samples were not fully digested after over-night incubation in the extraction buffer, additional Proteinase K (50 µg/mL) was added, and incubation continued for 5–8 h at 55 °C[57]. Between one and three different bone elements were sampled from each individual, and one to three extractions were performed per bone element. We processed one or two negative controls for every eight samples extracted.

Single-indexed blunt-end Illumina DNA libraries were prepared using P5 and P7 adapters following refs. [58] and [11] excluding the shearing step. We also prepared Uracil-DNA-glycosylase (UDG)-treated libraries, where post-mortem deaminated sites were cut with UDG and endonuclease VIII (endo VIII)[59] (see Supplementary Data 2 for library types per sample). Between one and five libraries were prepared from each DNA extract, and one negative library control was processed for every 6 to 8 ancient DNA libraries. We used quantitative PCR (qPCR) to determine the optimal number of PCR cycles for each library. Duplicate 25 µl qPCR reactions with 1 µl of DNA library, 1X Maxima SYBR Green Mastermix, and 200 nM of each IS7 and IS8 primers[58] were prepared and amplified following the supplier's instructions (ThermoFisher Scientific). Libraries were then amplified in two to four reactions for 12–18 cycles with approximately one negative control for every four reactions. Blunt-end libraries were prepared and amplified as in[11] and UDG-treated libraries were prepared and amplified as in[60] using IS4 and indices from[58]. Amplified libraries were quantified either on a TapeStation using a High Sensitivity kit (Agilent Technologies) or using a Bioanalyzer 2100 and a High Sensitivity DNA chip (Agilent Technologies). All DNA libraries were sequenced at SciLifes SNP & SEQ Technology platform in Uppsala, Sweden, using either Illumina HiSeq 2500 with v. 2 paired-end 125 bp chemistry or HiSeq X Ten with v. 2.5 paired-end 150 bp chemistry. The negative controls processed did not yield any DNA and were not sequenced.

**Processing of the raw sequence data and read mapping**. The obtained paired-end sequence reads were first merged (minimum required overlap was set to 11), and adapters following removed using AdapterRemoval v. 2.1.7[61] or MergeReadsFastQ_cc.py[62]. Trimmed and merged reads were mapped to human reference genome version hs37d5 using bwa aln Version: 0.7.17-r1188[63]. The following non-default parameter settings were used in the mapping: -l 16500 -n 0.01 -o 2[6,33]. PCR duplicates were detected and collapsed using a modified version of FilterUniqueSAMCons.py[62], including a random sampling of variant selection in duplicate collapsing. We excluded reads with reference identity less than 90 % or read length shorter than 35 bp.

**Basic statistics**. To evaluate each sequencing library's quality and quantity, we calculated the number of reads, the proportion of reads mapping to the human genome, average read length, clonality, and mean depth of coverage as in[11]. The genetic sex for each sample was determined based on the ratio of reads mapping to sex chromosomes[64] and X-to-autosome ratio[65]. For each specimen, we performed multiple rounds of sequencing. After an initial inspection of the data quality, we merged the mapped bam files from each library, including PCR duplicate removal, as described above. After the library merge, we combined all libraries from one individual into one bam file using samtools v. 1.5 merge option[66]. For each sampled individual, we inspected the read length distribution and signature of deamination at 5' and 3' read ends[67] using MapDamage v.2.0.8[68]. The read length and damage plots associated with the samples from this study are available from Figshare (https://doi.org/10.6084/m9.figshare.22811240.v1).

**Contamination estimation**. To estimate possible contamination in our sampled individuals, we utilized three different methods exploiting variation in haploid chromosomes. Two methods were based on mitochondrial (mt) reads[69,70] and were applied to all samples. X-chromosome-based contamination estimation was applied to male individuals as implemented in ANGDS v.0.921[71,72]. We used the same parameter settings as in the ANGSD manual except –minQ was set to 30.

**Mitochondrial haplogroup assignment**. We generated a consensus mitochondrial sequence based on the reads mapping to the hs37d5 mt reference assembly. For doing so we retained the most common allele on each sites and each individual with ANGSD[71] using the following command: angsd -i <inbam > -doFasta 2 -doCounts 1 -minQ 30 -minMapQ 30 -setMinDepth 3 -r MT: -out <outfasta > . Hence, the minimum number of bases per site was set to 3, minimum mapping, and base quality to 30. At polymorphic (heteroplasmic) sites, the most common base was retained.

The generated fasta files were used to call mitochondrial haplogroups using the standalone version of HaploGrep v. 2.1.16[73]. and the online version of HaploFind[74] with default settings. For each Haplofind assignment, we checked the missing

mutations and evaluated the assignment's reliability using PhyloTree Build 17[75]. If all the defining mutations of the haplogroup at the lowest assigned level were missing due to lack of data or if any of them had the ancestral allele, we manually assigned the haplogroup to a higher-level clade. We also checked the presence of additional previously defined variants if the HaploGrep and HaploFind assignments differed. The detected mt base substitutions are available from Supplementary Data 4.

**Y-chromosome analysis**. For individuals assigned as XY, we determined Y-chromosome haplogroups using the pipeline developed in[51]. In brief, the most likely haplogroups were assigned based on the genotype calls from 7773 trans-version no-indel branch determining polymorphisms obtained from the International Society of Genetic Genealogy collection (version 11.110 from April 21, 2016; https://isogg.org/). For each sample, we followed its Y-lineage based on derived mutations starting from the Y-chromosome tree root. The individuals were assigned to the lowest level haplogroup linked to the most likely lineage that did not show ancestral mutations. The derived mutations not connected to this lineage were not taken into consideration. Y-haplogroups were not assigned if the individual's mean genome depth was less than 0.1 X. The ancestral vs. derived state for accessible sites is available from Supplementary Data 5.

**Comparative dataset and reference panels**. In addition to the newly produced samples, we utilized a selected set of previously published Eurasian ancient individuals[6,8–12,14,17–22,29,30,33,35,36,51,52,76–83] as a comparative dataset. The full list of ancient individuals is available from Supplementary Data 3.

The ancient samples were overlapped with two different reference single nucleotide polymorphism (SNP) panels by randomly drawing a base from each SNP site on the given reference panel (so called pseudohaploid genotype calls). We used only transversions for blunt-end libraries, while for UDG-treated libraries, all sites were used. For half-UDG treated libraries, we clipped off five bp from both read ends using BamUtil v. 1.0.14 trimBam[84], and all sites were used in genotype calling. The alleles absent in the modern reference samples were coded as missing data for the specific ancient sample on the final dataset. The reference panels utilized in this study were (1) 594 024 autosomal SNP sites from the Affymetrix Human Origins fully public dataset[6,34]. (2) 1 938 919 autosomal transversions from the 1000 genomes project[85] of which minor allele frequency in the Yoruba (YRI) population were at least 0.1[86].

After the dataset overlap, we haploidized the modern samples to match the format of the ancient samples. For the overlap panels 2, we added chimpanzee genotypes from http://popgen.dk/software/download/angsd/hg19ancNoChr.fa.gz (downloaded September 8, 2020). Each position's genotypes were extracted using samtools faidx and written in tped format as homozygous for the reference allele. The dataset was converted to bed format and the genotype files were merged using Plink version 1.90b4.9[87]–bmerge option. Of the original sites, 29,347 were triallelic and were removed from the final combined dataset.

**Kinship**. We used program READ[37] to check for possible sample duplicates and kin pairs in our dataset. Since READ assumes no population structure within individuals explored, the following groups were analyzed in separate READ runs: (1) All Ukrainian samples from the current study and additional Mesolithic and Neolithic samples from nearby sites from Jones et al. [30] (samples StPet2, StPet12) and Mathieson et al. [12] (samples I6,561, I5,876, I5,885, I1,733, I1,734, I1,737, I1,763, I1,819, I3,717, I3,718, I4,111, I4,112, I4,114, I5,875, I5,881, I5,883, I5,886, I5,889, I5,890, I5,891, I5,892, I5,893). (2) All Romanian and Polish Mesolithic samples from this study in combination with relevant samples from González-Fortes et al. [22] (samples OC1, SC1, SC2) and Mathieson et al. [12] (samples I4,081, I4,582, I5,408, I4,607, I4,655, I5,411, I5,436, and I2,534). (3) Romanian Neolithic/Eneolithic samples from this study, and samples I2,532 and I2,533 from Mathieson et al. [12]. (4) The Polish Neolithic/Eneolithic samples from the current study combined with 11 age fitting samples from Fernandes et al. [36] (samples N18, N19, N20, N22, N25, N26, N27, N28, N31, N36, and N42).

We initiated the analysis by including all individuals. To confirm the detected kinship pairs, we rerun the analysis by excluding samples with little data (<0.1 X mean depth of coverage) and only including whole-genome sequence data. We used the 1000 genomes overlap panel for the kinship analysis except for the sample sets, including capture data, where we used the Human Origins overlap panel. We reported the kin pairs, whose standard error scaled normalized mean P0 score distance (Z-score) to the adjacent categories were >1.96.

**Population structure**. The general population structure of the investigated individuals was characterized using principal component analysis (PCA) implemented in EIGENSOFT smartpca[88,89]. The eigenvectors were estimated twice. First, using 768 individuals from 81 West Eurasian groups from the Human Origins dataset, and the ancient samples were projected onto the vector space estimated from the modern samples with the following settings:

altnormstyle: NO, numoutlieriter: 0, killr2: YES, r2thresh: 0.7, numoutlierevec: 0, lsqproject: YES, shrinkmode: YES

Additionally, we run model-based clustering of the 281 ancient individuals (Supplementary Data 3) and 43 West Eurasian, Central Asian & Siberian groups

(groups with at least ten individuals included only) from the Human Origins dataset using the software ADMIXTURE[31]. We first pruned the data for linkage disequilibrium using Plink version 1.90b4.9[87], and the following parameters–indep-pairwise 200 25 0.4 (parameters: SNP window size, step size, r2 threshold, respectively). We ran ADMIXTURE with a different number of clusters (k) ranging from 2 to 10, and for each number of clusters, we replicated the run ten times with different random starting points. For visualization, we used pong v. 1.4.7[90] to bundle together the membership coefficient matrices (Q) from different replicates and the different number of clusters.

**Conditional nucleotide diversity**. We calculated conditional nucleotide diversity (CND) following the procedure from ref. [33]. CND measures the number of pair-wise differences between pairs of pseudohaploid set of genotypes. Under the assumption that the combined haploid genomes are drawn from unrelated indi-viduals from the same population, CND reflects within-population diversity[33]. For CND calculation, we used the 1000 genomes SNP panel, and paired the individuals within the following groups, only including individuals of whose mean genome depth were at least 0.1 X: (1) Ukraine Mesolithic: Lower Dnipro Valley Epipa-leolithic/Mesolithic ukr125, ukr102, ukr108 (this study) and Ukraine_HG/StPet12[30]. (2) Ukraine Neolithic: Lower Dnipro Valley Neolithic/Eneolithic (this study: ukr005, ukr087, ukr111, ukr112, ukr113, ukr116, ukr117, ukr123, ukr144, ukr147, ukr158, ukr159, ukr160, ukr161, ukr162). (3) Sweden Mesolithic1: Huseby Klev Mesolithic ble004, ble008 from Kashuba et al. [80]. (4) Sweden Mesolithic2: Stora Förvar (Gotland) Mesolithic SF12 and SF9 from ref. [11]. (5) Sweden Meso-lithic3: Motala Mesolithic Motala-1, Motala-4, Motala-6, Motala-12[6]. (6) Southern Norway Mesolithic: Hummervikholmen Mesolithic Hum1 and Hum2 from Güther et al. [11]. (7) Romania Mesolithic: rom061 and rom066 from this study; SC1 & SC2 from ref. [22]. (8) Poland Mesolithic: poz297 and poz503 from this study. (9) Latvia Mesolithic: Hunter-gatherers from Zvejnieki ZVEJ25 & ZVEJ32[30]. (10= Romania Neolithic/Eneolithic: buk002, buk010, buk019, buk022, buk023, buk029, rom057_rom058, rom011, rom046, rom047 from this study. (11) Poland Neolithic/Eneolithic lbk102, lbk104, lbk138, poz120, poz177, poz236, poz252, poz275 from this study. We paired individuals in descending temporal order (except for kin pairs) to access the temporal patterns in diversity (sorted by median cal BP age) (Supplementary Data 7).

**F-statistics and admixture modeling**. To test the different models of shared ancestry and admixture in our study group, we calculated $f_3$-outgroup statistics (Yoruba; X, Y) and $f_4$-statistics (Chimpanzee/Yoruba, X; Y, Z) for multiple dif-ferent comparisons. A random sample of 20 Yoruban individuals were used in the $f_3$-statistics calculation. The f-statistics were calculated using AdmixTools v. 20160803[34] in an AdmixTools wrapper Admixr v. 0.7.1[91] in R v. 3.6.1[92]. To test for genetic continuity, we used the $f_3$-outgroup test and the highest coverage local Mesolithic individual as the group Y. Different admixture scenarios of the Meso-lithic and Neolithic groups were tested using qpAdm v. 401, and qpGraph v. 6100 from the AdmixTools package[34] with the following parameter settings:

qpWave & qpAdm: allsnps: YES, qpGraph: outpop: NULL, useallsnps: YES, blgsize: 0.05, forcezmode: YES, lsqmode: YES, diag: .0001, bigiter: 6, lambdascale: 1For each qpAdm set of models, an adjusted set of left and right groups were used to optimize the confidence interval estimation. See Supplementary Data for the model information.

We used f4-ratio[34] test ran in Admixr to estimate admixture proportions in the Neolithic dataset as follows: (1) The Polish Neolithic/Eneolithic dataset: f4ratio(X = Poland_Neolithic, A = poz503, B = poz297, C = Bar8, O = Chimpanzee). (2) The Romanian Neolithic/Eneolithic dataset: f4ratio(X = Romania_Neolithic, A = OC1, B = SC2, C = Bar8, O = Chimpanzee).

Covertf program from the AdmixTools v. 20160803[34] package was used throughout the study to covert the data from plink to eigenstrat format.

**Test for genetic continuity**. Continuity through time was tested using the Anchor Method[32]. In short, we condition upon heterozygous ancestral/derived sites in a given so called anchor individual and count the proportion of derived alleles at those sites in other test individuals. The proportion derived at conditioned sites is expected to remain constant forwards in time along the branches of a phylogeny, but reduces backwards in time, providing a signal to detect population continuity and ghost admixture among a heterochronous sample (see McKenna et al. [32]. for details). All-site vcfs were used as input to the Anchor method (details of the processing and diploid genotype calling are found in[93]). Analyses used all sites for UDG-treated sequence data and were restricted to transversion sites for non-UDG treated sequence data.

**Route optimization via cost surface analysis**. We used Geographic Information System-based path distance and least-cost path (LCP) computation to determine the minimum accumulative travel cost from five different WHG sites (Bichon, Ranchot88, Rochedane, Loschbour, Villabruna, coordinates obtained from ref. [8]) to the selected Mesolithic sites. To calculate the path distance from the start sites, we created cost raster and surface raster as an input variable for the analysis[94]. Cost raster represents the cost-per-unit for moving through the cell, and surface raster the difficulty of moving from one cell center to another. Present Global Land Cover[95] was used to produce a cost to travel across land and water areas travel in the Western Eurasia region with a resolution of 30 arc seconds. We used the percentage values of water coverage in each cell of the land cover type. Original percentage values have been reclassified into six cost value classes on a scale between 1 to 16. The cells with 0% water cover were reclassified to one, 1–20% to 2, 21–40% to 4, 41–60% to 6, 61–80% to 8, and 81–100% to 16. Values were assigned regarding the difficulty of moving across each land cover classes. Land areas were assumed the easiest for traveling and was therefore assigned a value one. As the percentage of water in the raster cell increase, we assumed that it would increase the friction of the movement. High percentages indicate large water bodies and wetlands. Noteworthy, the water cover does not restrain the traveling fully but it only increases the friction to cross the landscape, and the added cost is low if the water percentage is low. In this study, we assumed that points with high water percentage (e.g., offshore areas) acted as isolation barrier even though probably occasionally crossed. Waterways are commonly assumed to be the main routes during prehistoric times. In our analysis, this is also often shown to be the case as rivers and inland lakes become the preferred route since the slope along them is shallower than other areas. In addition, to consider the areas covered by the continental ice sheet 10,000 years ago[96,97], we reclassified the original data by adding a value of 999 for those areas assuming that people would not be willing to travel across ice sheets. Overall, we assumed that people more apparently choose the easier landscape to cross before using the most difficult one. The final cost raster is created by adding together the reclassified datasets.

Global Multi-resolution Terrain Elevation Data[98] was used as a surface raster. Elevation values define the vertical difficulty encountered in moving from one cell to the next. We used Inverse Linear parameter of a vertical factor modifier of ArcGIS pro 2.7.0 Path distance tool. The vertical relative moving angle (VRMA) was set to 5 degrees, meaning that VRMA below or above 5° was set to infinity in the analysis. The Inverse Linear parameter favors movement gently sloping downhill. Path distance analysis produces a distance raster that contains information on the accumulative cost over a cost surface while compensating the actual surface distance that need to be traveled and for the vertical factors influencing the total cost of moving from one site to another. Analysis also produces backlink raster that identifies in which cell to move into on its way back to the source site. LCP was carried out with Cost path tool in ArcGIS pro using both distance and backlink raster.

The least-cost routes obtained from this analysis were used to measure the geographical accessibility of a given point from the specified start points to test the relationship of genetic and spatial distance between datapoints. Hence, they should not be interpreted as suggested migratory routes per se. Furthermore, we acknowledge that there are several uncertainties in the accessibility of a given site, such as glaciation and sea-level variation in Northern Europe.

**Linear regression analysis**. We used linear regression analysis to investigate the dependence of admixture proportions in the Mesolithic East and North European individuals relative to geography distance between sites (admixture isolation-by-distance). The admixture isolation-by-distance model assumes that the proportion of contribution from each source population is inversely correlated with the geo-graphic distance from the source population. Based on the descriptive admixture results (PCA & Admixture), we assumed two source populations closely related to two lineages; Western Hunter-Gatherers and Paleolithic Siberians (represented by AfontovaGora3). In a simple balanced isolation-by-distance admixture zone, the genetic contribution from a population A is assumed to be 100% at the site of the source populations while at this site the genetic contribution from the population B is assumed to be 0%. In the geographic midpoint, the admixture proportions from each source group are expected to be at 50%.

To test whether geographic distance explains the admixture proportions in the Mesolithic samples from Northern and Eastern Europe, the shortest least cost route from the WHG core region and AfontovaGora3 were used as explanatory variables for the admixture measure f4(Chimp, X; AfontovaGora3, Loschbour). To avoid possible batch effects on our $f_4$ calculations, we did not include individuals investigated by SNP-capture method in our set of X individuals. Additionally, we excluded individuals of which f4 calculations were based on less than 5000 sites. We fitted linear model y ~ ax +bz + c to the dataset using lm function in R v. 3.6.2[92]. The lm diagnostic plots were inspected to detect highly influential points and evaluate if the dataset met the critical model assumptions.

**Reporting summary**. Further information on research design is available in the Nature Portfolio Reporting Summary linked to this article.

## Data availability
The sequence data generated in this study is available at the European Nucleotide Archive under the project PRJEB59598. A collection of read length and damage plots; numerical source data for graphs; auxiliary scripts, and R notebooks for plotting and analysis generated in this study are available from Figshare (https://figshare.com/projects/Data_and_scripts_from_Mattila_et_al_2023_Genetic_continuity_isolation_and_gene_flow_in_Stone_Age_Central_and_Eastern_Europe/167072). The land cover and elevation raster data used in this study are available from FAO map catalog

(http://www.fao.org/geonetwork/srv/en/main.home?uuid=ba4526fd-cdbf-4028-a1bd-5a559c4bff38) and USGS (https://topotools.cr.usgs.gov/gmted_viewer/viewer.htm). The glacial information is available from PANGAEA information system (https://doi.org/10.1594/PANGAEA.848117). All the scripts for running the Anchor method and plotting results are available from https://github.com/jammc313/Genetic-continuity.git.

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

## Acknowledgements

We thank Hanna Edlund for laboratory assistance, Federico Sanches-Quinto for advice related to the comparative genetic datasets, Julia Koelman for reviewing the family trees presented in this study, and Harri Antikainen for guidance in the route optimization and cost surface analysis. Malcolm Lillie is acknowledged for discussing the archeology of the Ukrainian dataset. Alexandra Comșa, Mircea Răchită, and Angela Simalcsik are acknowledged for providing samples and sample information from some of the Romanian specimens. Sequencing was performed by The National Genomics Infrastructure (NGI) Uppsala. Uppsala Multidisciplinary Center for Advanced Computational Science (UPPMAX) provided data storage and computational resources for this project (projects snic2018-8-304, snic2019-8-270, snic2020-2-10 & snic2021-22-823). This work was funded by the Finnish Cultural Foundation (to T.M.M.), Wenner-Gren Foundations (projects UPD2018-0308 and P2020-0008 to T.M.M. under supervision of M.J.), and by the Knut and Alice Wallenberg Foundation (to M.J., A.G., and J.S.). H.M. was supported by Swedish Research Council grant no. 2017-02503 and by Riksbankens Jubileumsfond grant no. P21-0266.

## Author contributions

E.M.S., H.M., A.J., M.N., M.J., A.G., A.N., and J.S. formulated the study and the sampling setup. Mi.C., I.P., M.R., A.S., Ł.P., S.W., K.S., P.M., L.C., M.S., J.K., and J.Ł. provided samples. A.J., Ma.C., H.M., and E.M.S. coordinated the radiocarbon dating, DNA extraction, library preparation, and sequencing of the material. T.M.M., H.M., N.K., P.P., T.G., and M.J. designed the genetic analyses. T.M.M. and T.G. developed computational pipelines. T.M.M., T.G., and J.M. analyzed the sequence data. T.M.M. and T.A.-H. designed and performed the cost-surface and route optimization analyses. N.P., M.R., H.M., N.K., P.P., I.P., A.N., M.R., Mi.C., Ma.C., A.J., Ł.P., and S.W. investigated the material's archeological context and wrote the archeological description of the dataset. T.M.M. wrote the manuscript with contributions from H.M., T.A.-H., N.K., P.P., A.N., Ł.P., Ma.C., A.J., Mi.C., J.M., and M.J. All the authors have seen and accepted the final version of the manuscript.

## Funding

## Competing interests

The authors declare no competing interests.
