## [Peer Review File · Communications Biology]

Reviewers' comments:

Reviewer #1 (Remarks to the Author):

Mattila et al. generated 56 ancient whole-genomes (0.01-4.55X) of the Mesolithic, Neolithic, and Eneolithic individuals from the Central and Eastern Europe, where hunting-gathering lifeways are known to have been maintained longer than those in the Southern and Western Europe. The temporal and spatial coverage includes Romania, Poland, and the lower Dnipro Valley region in Ukraine over a period spanning from 10,500 to 5,500 years BP. This set of high-quality ancient genomes allows the authors to address several key questions: genetic continuity during the cultural shift from foraging to farming, origins of Eastern hunter-gatherers, impacts of Anatolian farmers on the genetic makeup of Eastern Europeans, and kinship in Stone Age Europe. Most of the results are convincing and support the conclusions made in this study. I really enjoyed reading this paper and don't see any problem in their statistical analysis. I have some minor comments below.

1. The values of f_4 -statistics were used in the linear regression analysis (Fig. 4b). But, I suppose these estimates come with variances, which can be large in particular if the depth of sequence coverage is low. Can the authors still see the same pattern even if those values are replaced with WHG ancestry estimated from qpAdm under two-admixture models (AfontovaGora3 and Loschbour)? The authors should calculate nested p-values to see if two-admixture is always better fit to the data than single-ancestry models. Also how did the authors define the core region of WHG? How are the results robust against the choice of this core region?

2. Genetic continuity tested with outgroup- f_3 statistics should be validated by qpAdm with a single-ancestry model (i.e., no admixture model) and a method developed in the paper ("Assessing the Relationship of Ancient and Modern Populations", Schraiber 2018 *Genetics* 208(1):383-398) as the outgroup- f_3 is just quantifying shared genetic drift between individuals or populations, rather than directly testing genetic continuity.

3. Some of the figures are not easy to see, and please see my suggestions or comments below:

- Fig. 2b: I am not entirely sure but think those newly sequenced individuals are labelled with their unique IDs, which are not so easily recognisable about where they come from and how old they are. Can the authors label those individuals by geographic and archaeological contexts as is the case for "Ukraine Mesolithic" or "Poland Neolithic" for example?

- Fig. 3: Again, those individuals IDs are not easy to recognise. Can the authors add geographic and archaeological information into these plots?

- Fig. 4c: The shapes in the legend are not visible. Can the authors remove lines overlaid with the shapes in the legend?

- In the main text, Fig. 4A seems to be mislabelled with Fig. 5A (e.g., Page 11, Line 212).

- Fig. S5.3: What error bars represent?

Reviewer #2 (Remarks to the Author):

Brief summary of the manuscript

The submitted paper evaluates human migrations during the Mesolithic-Neolithic transition in Europe using ancient genomic data, and reveals the admixture history of hunter-gatherers in Eastern Europe over two periods using a new data set. In addition, kinship analysis attempts to shed light on marriage patterns and sociality at the time.

Overall impression of the work

I have the impression that the setting of the research theme, the method of analysis of ancient

geneomic data, the method of describing the results, and the details of the discussion are all very well presented. I highly recommend the publication of this paper, as it has reached a level comparable to other Nature journals.

Specific comments, with recommendations for addressing each comment
Please make a minor revision.

On page 5, lines 117 and 119, the word "BP" should be changed to "calBP" as elsewhere.
Also, for the PCA in Figure 2A, please add the regional name near AfontovaGora3 and WHG, respectively, and add the name of the region and period near Yamnaya. This is for easy understanding for the general reader.

Reviewer #3 (Remarks to the Author):

In the present study of Mattila et al., the genome-wide sequencing data of 56 ancient hunter-gatherer and early farmer individuals from Stone Age Central and Eastern Europe were obtained. The authors characterized the genomic diversity of ancient individuals from this region, assessing their genomic affinities with other ancient groups from Europe and Asia.

The manuscript by Mattila and co-authors represents a breakthrough for the regional history of Central and Eastern Europe through the generation of novel genomic information, as well as chronological and isotopic data, from Mesolithic and Neolithic human remains from Poland, Romania and Ukraine, in archaeological contexts somewhat underrepresented in paleogenomic research in Europe.

The contextualization of Stone Age Europe is clear, as are the objectives. The methodological approach to fulfill them is adequate and the data and results are well presented. Figures are attractive and necessary. Inferences and conclusions are not speculative, clearly arising from genomic data in relation to archaeological contexts.

However, I have some comments, such as the following:

1) As a reviewer specializing in the study of peopling and evolutionary history through paleogenomics of another continent, I know the general but not the details of the archaeological/paleogenomic context of Stone Age Europe. So I cannot realize if there are missing groups/individuals that were not included in the analyses, nor comparisons that should have been done. In this regard, I would appreciate (as, I imagine, would other non-expert readers of CommsBio) a brief mention of the archaeological context/location the individuals that are used as obvious references for specific temporal or cultural contexts (i.e., Afontova Gora3, Sidelkino, Loschbour). Afontova Gora3 and Loschbour were included in the map of Fig. S5:5 (I noted it for inclusion in the main text) but Sidelkino is not on any map (perhaps the brief description would be enough).

2) One of my concerns, in general, of paleogenomic works is that they are not easily accessible to all readers, even to archeologists who are potentially the most interested in the conclusions reached. So I emphasize making the data and results the most transparent and understandable as possible. I do not expect an explanation of how to calculate f_3 and f_4 statistics, but to relate the values obtained to the inferences that can be made. For example, in discussing population continuity (lines 144-150) you state: "The individuals from the Neolithic lower Dnipro Valley were genetically very similar to the Epipaleolithic/Mesolithic individuals ... the Romanian and Polish sampling sites ... were genetically similar to the Anatolian Neolithic farmers. These results were also supported by the patterns of allele sharing with WHG and EN (Fig. 3A-C) as well as the uniparental markers..." In this case, I suggest including something that clarifies the figure: "These results were also supported by the patterns of

allele sharing with WHG and EN (Fig. 3A-C), where negative values mean continuity/affinity/whatever with EN... and positive values are interpreted as continuity with WHG...". The same applies to the Figures 3D-F.

Other examples, but not limited to these, that could be improved in this regard:

- Lines 199-201: "The f_4 -values were negative for all, but for the Polish individuals, they were not significantly different from 0 (Supplementary Fig. S5:3A), which means xxx."

-Lines 278-280: "Other ancient neighboring groups AN, CHG, EHG, Neolithic Iranian WC1, Mal'ta, WHG and Sunghir were used as 'right' populations in addition to an outgroup (chimp, Supplementary Dataset S10)." Explain what a 'right' population is.

3) Conditional Nucleotide Diversity (CND) was presented in two sections: "Over 4,000 years of genetic continuity...", to test diversity as an indicator of population size and continuity without gene flow, and "Isolation-by-distance..." to test ancestry proportions. But in the latter it appears as a new or different analysis, whereas it was already described. Either the paragraphs are unified or the above analysis is taken up again.

4) It would be illuminating if you could include a brief description of the burial contexts of the Romanian and Ukrainian families (distance between skeletons, presence of unrelated individuals in the same grave, etc.

5) There is no ethical statement regarding work with ancient humans. It is necessary to mention in Materials and Methods whether the work has the endorsement of an Ethics Committee. It is my understanding that in Europe permits from National or Regional institutions or Cultural Organizations are not required, nor is the consent of native communities or other stakeholders to carry out bioanthropological and/or genomic studies, as in other regions of the world. But work is still being done with human remains, so ethical issues must be taken into account.

6) Here are some additional comments:

- The manuscript is very well written and easy to read.

- Other comments and edits to the main text are detailed in the attached pdf file.

- The supplementary information file was also reviewed, and some comments and edits can be found, but the archaeological context and ancient individuals were not thoroughly reviewed to detect and point out minor errors.

Responses to Reviewer's comments:

Reviewer 1:

1. The values of f_4 -statistics were used in the linear regression analysis (Fig. 4b). But, I suppose these estimates come with variances, which can be large in particular if the depth of sequence coverage is low. Can the authors still see the same pattern even if those values are replaced with WHG ancestry estimated from qpAdm under two-admixture models (AfontovaGora3 and Loschbour)? The authors should calculate nested p-values to see if two-admixture is always better fit to the data than single-ancestry models. Also how did the authors define the core region of WHG? How are the results robust against the choice of this core region?

Author response: We have added a short description on how the WHG core region was approximated (lines 249-250, see also Methods from line 657 ->) and the nested p-values to the Dataset S6. The analysis was also repeated using two different methods approximating the geographic distance from WHG; 1. median distance among the five included WHG sites and 2. a sum of all five WHG distances. In addition, similar results were also obtained if the f_4 -values were replaced with the WHG ancestry estimated with qpAdm (added text lines 261-263 & Supplementary Data S8). As mentioned in the main text (lines 213-214), qpAdm analysis was not able to separate the admixture model in all cases but this is mostly due to power issues since samples with low coverage tend to show the lack of statistical support. However, the f_4 -statistics shows significant increase in allele sharing in the tests $f_4(\text{chimp};X;\text{WHG},\text{EHG})$ in samples from all sites when we use the 1000 genomes overlap panel (Supplementary Fig S5:4) which confirms the allele sharing with the eastern lineage. In this comparison we were not able to use AG3 since it is capture data and not WGS but given the other results, we concluded significant evidence for admixture in most of the Mesolithic cline samples.

2. Genetic continuity tested with outgroup- f_3 statistics should be validated by qpAdm with a single-ancestry model (i.e., no admixture model) and a method developed in the paper ("Assessing the Relationship of Ancient and Modern Populations", Schraiber 2018 *Genetics* 208(1):383-398) as the outgroup- f_3 is just quantifying shared genetic drift between individuals or populations, rather than directly testing genetic continuity.

Author response: We have validated our results using a new method designed for testing genetic continuity (McKenna et al. 2022 <https://www.biorxiv.org/content/10.1101/2022.12.01.518676v1.abstract>). See lines 154-156, 646-655, Supplementary Fig. S5:3

3. Some of the figures are not easy to see, and please see my suggestions or comments below:

- Fig. 2b: I am not entirely sure but think those newly sequenced individuals are labelled with their unique IDs, which are not so easily recognisable about where they come from and how old they are. Can the authors label those individuals by geographic and archaeological contexts as is the case for "Ukraine Mesolithic" or "Poland Neolithic" for example?

Author response: Following the reviewer's suggestion, we have added an additional geographic and time period label for the newly sequenced individuals, and shaded them to make the figure easier to read.

- Fig. 3: Again, those individuals IDs are not easy to recognise. Can the authors add geographic and archaeological information into these plots?

Author response: We have added regional and time period indicators following the reviewer's suggestion

- Fig. 4c: The shapes in the legend are not visible. Can the authors remove lines overlaid with the shapes in the legend?

Author response: The lines were removed as suggested

- In the main text, Fig. 4A seems to be mislabelled with Fig. 5A (e.g., Page 11, Line 212).

Author response: Corrected (see line 219)

- Fig. S5.3: What error bars represent?

Author response: Description added (see Fig. S5:4, note updated figure numbering)

Reviewer 2:

1. On page 5, lines 117 and 119, the word "BP" should be changed to "calBP" as elsewhere.

Author response: Corrected (see lines 115 & 117)

2. Also, for the PCA in Figure 2A, please add the regional name near AfontovaGora3 and WHG, respectively, and add the name of the region and period near Yamnaya. This is for easy understanding for the general reader.

Author response: Geographical and period labels have been added in general to ease the reading of the plot (see Fig. 2A)

Reviewer 3:

1) As a reviewer specializing in the study of peopling and evolutionary history through paleogenomics of another continent, I know the general but not the details of the archaeological/paleogenomic context of Stone Age Europe. So I cannot realize if there are missing groups/individuals that were not included in the analyses, nor comparisons that should have been done. In this regard, I would appreciate (as, I imagine, would other non-expert readers of CommsBio) a brief mention of the archaeological context/location the individuals that are used as obvious references for specific temporal or cultural contexts (i.e., Afontova Gora3, Sidelkino, Loschbour). Afontova Gora3 and Loschbour were included in the map of Fig. S5:5 (I noted it for inclusion in the main text) but Sidelkino is not on any map (perhaps the brief description would be enough).

Author response: The different genetic groups were described in the introduction to get an overall picture of the spatiotemporal patterns of genetic variation in the study region, and the reference individuals were linked to each group when first time discussed. We have now added some additional description in the text and figure legends (e.g., updated Fig. 2).

2) One of my concerns, in general, of paleogenomic works is that they are not easily accessible to all readers, even to archeologists who are potentially the most interested in the conclusions reached. So I emphasize making the data and results the most transparent and understandable as possible. I do not expect an explanation of how to calculate f_3 and f_4 statistics, but to relate the values obtained to the inferences that can be made. For example, in discussing population continuity (lines 144-150) you state: “The individuals from the Neolithic lower Dnipro Valley were genetically very similar to the Epipaleolithic/Mesolithic individuals ... the Romanian and Polish sampling sites ... were genetically similar to the Anatolian Neolithic farmers. These results were also supported by the patterns of allele sharing with WHG and EN (Fig. 3A-C) as well as the uniparental markers...” In this case, I suggest including something that clarifies the figure: “These results were also supported by the patterns of allele sharing with WHG and EN (Fig. 3A-C), where negative values mean continuity/affinity/whatever with EN... and positive values are interpreted as continuity with WHG...”. The same applies to the Figures 3D-F. Other examples, but not limited to these, that could be improved in this regard: - Lines 199-201: “The f_4 -values were negative for all, but for the Polish individuals, they were not significantly different from 0 (Supplementary Fig. S5:3A), which means xxx.” -Lines 278-280: “Other ancient neighboring groups AN, CHG, EHG, Neolithic Iranian WC1, Mal’ta, WHG and Sunghir were used as ‘right’ populations in addition to an outgroup (chimp, Supplementary Dataset S10).” Explain what a ‘right’ population is.

Author response: We have generally modified the writing to improve the clarity of conclusions drawn. See lines 146-150, 203-212,219, for examples pointed out by the reviewer.

3) Conditional Nucleotide Diversity (CND) was presented in two sections: "Over 4,000 years of genetic continuity...", to test diversity as an indicator of population size and continuity without gene flow, and "Isolation-by-distance..." to test ancestry proportions. But in the latter it appears as a new or different analysis, whereas it was already described. Either the paragraphs are unified or the above analysis is taken up again.

Author response: We used conditional nucleotide diversity to evaluate two differing models (1. population size through time and 2. population size in space) with only partially overlapping data and hence these results were discussed in two different occasions. For the sake of flow of the text, we have kept these parts separate. We have modified the text to make this separation more clear (lines 164-172 & 297-304).

4) It would be illuminating if you could include a brief description of the burial contexts of the Romanian and Ukrainian families (distance between skeletons, presence of unrelated individuals in the same grave, etc).

Author response: We have included additional details for the Ukrainian kin (see lines 339-368). Unfortunately, the information for the Romanian kin trio was sparse since it was a rescue excavation.

5) There is no ethical statement regarding work with ancient humans. It is necessary to mention in Materials and Methods whether the work has the endorsement of an Ethics Committee. It is my understanding that in Europe permits from National or Regional institutions or Cultural Organizations are not required, nor is the consent of native communities or other stakeholders to carry out bioanthropological and/or genomic studies, as in other regions of the world. But work is still being done with human remains, so ethical issues must be taken into account.

Author response: Ethical statement added into the methods section (see lines 419-422)

6) Here are some additional comments:

- The manuscript is very well written and easy to read.
- Other comments and edits to the main text are detailed in the attached pdf file.
- The supplementary information file was also reviewed, and some comments and edits can be found, but the archaeological context and ancient individuals were not thoroughly reviewed to detect and point out minor errors.

Author response: The minor edits have been included in the revised version of the manuscript

REVIEWERS' COMMENTS:

Reviewer #1 (Remarks to the Author):

I am satisfied with the responses made by the authors. There is no ant further comment on the manuscript.

Reviewer #3 (Remarks to the Author):

After reading the revised version of the document "Genetic continuity, isolation, and gene flow in Stone Age Central and Eastern Europe" by Mattila and co-authors, I am glad to see and very much appreciate that the authors have responded to all of my suggestions and those of the Reviewers 1 and 2. I consider this version suitable for publishing in Communications Biology as is.